# Molecular screening for the mutation associated with canine degenerative myelopathy (*SOD1:c.118G > A*) in German Shepherd dogs in Brazil

**Cássia Regina Oliveira Santos**[1,2]*, **João José de Simoni Gouveia**[3], **Gisele Veneroni Gouveia**[3], **Flávia Caroline Moreira Bezerra**[1], **Joel Fonseca Nogueira**[1], **Durval Baraúna Júnior**[4]

1 Postgraduate Program in Veterinary Sciences in the Semiarid, Federal University of Vale do São Francisco, Petrolina, Pernambuco, Brazil, 2 University Veterinary Clinic, Federal University of Vale do São Francisco, Petrolina, Pernambuco, Brazil, 3 Department of Animal Sciences, Federal University of Vale do São Francisco, Pernambuco, Brazil, 4 Department of Veterinary Medicine, Federal University of Vale do São Francisco, Petrolina, Pernambuco, Brazil

* cassia.santos@univasf.edu.br

**Data Availability Statement:** All relevant data are within the manuscript and its Supporting information files.

## Abstract

Canine Degenerative Myelopathy is a late onset recessive autosomal disease characterized by a progressive ascending degeneration of the spinal cord. Two causal mutations are associated with this disease: a transition (*c.118G>A*) in exon 2 of the *SOD1* that was described in several breeds and a transversion (*c.52A>T*) in exon 1 of the same gene described in Bernese Mountain dogs. The aim of this study was to understand the impact of the *SOD1: c.118G > A* mutation by genotyping a population of German Shepherd dogs in Brazil. A PCR-RFLP approach was used to genotype 97 healthy individuals belonging from the Northeast (Bahia and Pernambuco states) and South (Santa Catarina state) regions of Brazil. A total of 95 individuals were successfully genotyped resulting in an observed genotype frequency (with 95% confidence interval) of: 0.758 (0.672–0.844), 0.242 (0.156–0.328) and 0.000 (0.000–0.000) for "*GG*", "*AG*" and "*AA*" genotypes, respectively. To our knowledge, this is the first attempt to describe the presence of the "*A*" allele associated with CDM (*SOD1:c.118G > A*) in German Shepherd dogs in Brazil and, as such, these results contribute toward important epidemiological data in this country and to the knowledge of the distribution of the aforementioned mutation worldwide.

## Introduction

Canine degenerative myelopathy (CDM) is characterized by a progressive ascending degeneration of the spinal cord. It was first described in the German Shepherd dog; however, it has been reported in other breeds, such as American Eskimo dog, Bernese Mountain dog, Boxer, Cardigan or Pembroke Welsh Corgi, Chesapeake Bay Retriever, Golden Retriever, Kerry Blue Terrier, Poodle, Pug, Rhodesian Ridgeback, and Siberian Husky [1–3].

**Funding:** The authors received no specific funding for this work.

**Competing interests:** The authors have declared that no competing interests exist.

Age of onset of neurologic signs is usually five years of age or older, with a mean of nine years in large breed dogs, but this varies among affected breeds [4]. Owners usually opt for euthanasia as paralysis ascends or when there are signs of respiratory or urinary dysfunction [4].

Diagnosis is based on the clinical characteristics of the disease and exclusion of other diseases with similar symptoms. A definitive diagnosis requires confirmation via histopathology [3]. Histopathology findings are compatible with a central axonopathy of the spinal cord. Axonal and myelinic degeneration affect all the funiculi, but may affect mainly lateral and dorsal funiculi. Neuronal cell body loss is also observed [4, 5].

From a genetic point of view, CDM can be characterized as a recessive autosomal disease with incomplete penetrance. Two candidate causal mutations are described. The first one, reported in Boxers, Rhodesian Ridgebacks, German Shepherds, Pembroke Welsh Corgis, and Chesapeake Bay Retrievers is a transition (*c.118G>A*) in exon 2 of the *SOD1* gene leading to a change from glutamate to lysine in amino acid 40 of the polypeptide chain [6]. A second mutation was reported in Bernese Mountain dogs, where the transversion (*c.52A>T*) located in exon 1 of the same gene (resulting in a change from threonine to serine at position 18 of the polypeptide chain) was also associated with manifestation of the disease [7, 8].

Additionally, a haplotype was described as a modifier related to the time of onset of this disease. Variations in the genetic transcription mediated by the SP110 (nuclear body protein) haplotype observed in Pembroke Welsh Corgi dogs with a mutation in *SOD1*: *c.118G > A* could be associated, at least in part, with the varying risk of developing CDM at a younger age [9].

Epidemiological information on genetic diseases in each breed are essential for establishing prevention plans [10]. Thus, epidemiological surveys must be performed in each country and region, and these results published at a global level, so that the allele distribution for a certain disease can be known [11].

It has been suggested that knowledge of the allele frequencies for *SOD1*: *c.118G>* in canine breeds predisposed to CDM may be used for selecting better strategies in genetic improvement programs [6]. Allele frequencies that were considered to be relatively high have been reported in German Shepherds (estimated at 0.35), which may already present a problem in terms of an extremely rigorous restriction in breeding carrier individuals [12].

In Boxer and Pembroke Welsh Corgi breeds, where higher allele frequencies have been reported (greater than 0.7) [6], a radical restriction in breeding carriers of the mutant gene will likely generate other consequences, such as the elimination of desirable characteristics or the unintentional selection of other mutations [12].

To the best of our knowledge, this is the first study aiming to screen for the mutation associated with CDM (*SOD1:c.118G > A*) in Brazil. As such, our study aims to contribute to the knowledge of the distribution of the aforementioned mutation worldwide and to investigate aspects involved in dissemination probability of CDM using a population of German Shepherd in Brazil.

## Material and methods

### Samples and DNA extraction

The present study was authorized by the Ethics Committee for the Use of Animals from the Federal University of Vale do São Francisco (CEUA/UNIVASF) under authorization number 0005/140217. All samples were collected and analyses performed after obtaining written informed consent from owners.

**Table 1. Gender, age, and location of the German Shepherds enrolled in this study.**

| Breeding stock/Owner | Location | N | Age (years) |
|---|---|---|---|
| 1 | Santa Catarina | 56 (25M;31F) | 1.5–11 |
| 2 | Bahia | 16 (12M;4F) | 2.5–7 |
| 3 | Bahia | 16 (3M;13F) | 5–11 |
| 4 | Bahia | 6 (6M) | 1–5 |
| * | Pernambuco | 3 (2M;1F) | 7–13 |

*Individuals enrolled during medical appointment in the Federal University of Vale do São Francisco Veterinary Clinic; M = Male; F = Female.

Ninety-seven (n = 97) German Shepherd dogs from the Northeast (Bahia and Pernambuco states) and South (Santa Catarina state) regions of Brazil were used. From these, 94 dogs belonged to four breeding stocks and were healthy at the time of sample collection. Three dogs were enrolled after presenting to the Federal University of Vale do São Francisco Veterinary Clinic for issues unrelated to CDM (Table 1). Signalment, location and genotype of enrolled dogs is presented in S1 Table.

No complaints related to clinical symptoms associated with CDM were reported by the owners of the studied animals. Samples were collected using oral swabs, and DNA was extracted using an in-house salting out protocol [13].

Additionally, a DNA sample from a Swiss Shepherd dog previously confirmed as homozygous (*A/A*) for the CDM-associated mutation (*SOD1:c.118G > A*) and with clinical and pathological signs compatible with the studied disease [14] was used as a control for the genotyping procedure.

## Investigation of the presence of the *c.118G>A* mutation in exon 2 of the *SOD1* gene

Genotyping of the *c.118G>A* mutation in exon 2 of the *SOD1* gene was performed using polymerase chain reaction-restriction fragment length polymorphism (PCR-RFLP). For partial gene amplification, primers DM_F (5′–AGTGGGCCTGTTGTGGTATC–3′) and DM_R (5′–TCTTCCCTTTCCTTTCCACA–3′) were used [12]. The enzyme TopTaq DNA Polymerase (QIAGEN) was used for the reactions following the manufacturer's recommendations. Reactions occurred in a final volume of 50μL with initial denaturation at 95°C for 10 minutes, followed by 40 cycles of denaturation at 94°C for 30 seconds, annealing at 53.2°C for 30 seconds, and extension at 72°C for 1 minute, followed by a final extension at 72°C for 10 minutes. Amplification products were detected after electrophoreses in 2% agarose gel stained with GelRed (Biotium).

Polymerase chain reaction products were digested using ACUI enzyme (BioLabs) under the following conditions: 5μL of digestion buffer, 0.57μL of SAM (S-adenosyl methionine), 23.43μL of ultrapure water, 2μL of ACUI enzyme, and 20μL of PCR product, totalizing 50μL. Reactions were incubated at 37°C for 6 hours, followed by inactivation of the enzyme at 65°C for 20 minutes. To determine the genotype of the dogs, digestion products underwent electrophoresis using agarose gel at 3% stained with GelRed (Biotium).

It is expected that PCR produces an amplicon with 292bp. After digestion, homozygous individuals for the normal allele (*GG*) are expected to produce two fragments (230bp and 62bp), heterozygous individuals (*AG*) are expected to produce three fragments (292bp, 230bp, and 62bp), and homozygous dogs with the mutation associated with CDM (*AA*) are expected to produce only one fragment of 292bp.

### Validation of genotyping via Sanger sequencing

Seven dogs were selected for sequencing (one with *AG* genotype, three with *GG* genotype, and three that could not be genotyped by PCR-RFLP). Additionally, the electropherogram of the Swiss Shepherd dog previously confirmed as *AA* homozygous was used as a reference for SNP analysis.

Partial amplification of the *SOD1* gene using primers F1_SOD 5′-GTCCCCAGCCTAGAAT GGTTAA-3′ and R2_SOD 5′-CGGCTTTGTGGATCATTTCC-3′ [15] was performed using TopTaq DNA Polymerase (QIAGEN) enzyme. The reactions had a final volume of 50μL, with initial denaturation at 95˚C for 10 minutes, followed by 40 cycles of denaturation at 95˚C for 30 seconds, annealing at 54˚C for 30 seconds, and extension at 72˚C for 1 minute, followed by a final extension at 72˚C for 10 minutes. The PCR products were sent for unidirectional Sanger sequencing at a private company.

Quality analysis and visualization of the resulting sequences were done using Phred Phrap Consed software suite [16, 17]. Good quality sequences were used to validate the genotypes using PolyPhred software [18], which were confirmed via visual inspection of the electropherograms and comparison to the reference sequence for the canine *SOD1* gene in the Ensembl database (ENSCAFG00000008859). The sequences were also analyzed using Poly Peak Parser and ClustalW, the latter available in MEGA7 software [19–21]. Figures of the electropherograms were generated using CLC Genomics Workbench 20.0 (QIAGEN).

Statistical analysis of allele frequencies was performed using HW_TEST software [22].

## Results and discussion

As expected, PCR produced an amplicon with 292bp. After ACUI digestion, homozygous individuals for the normal allele (*GG*) presented two fragments (230bp and 62bp), heterozygous individuals (*AG*) presented three fragments (292bp, 230bp, and 62bp) and homozygous dogs with the mutation associated with CDM (*AA*) presented only one undigested fragment of 292bp. A weak band corresponding to partial digestion of the PCR products could be seen in homozygous dogs (*GG*) at 292pb, but did not compromise interpretation of results (Fig 1).

From the ninety-eight individuals genotyped using PCR-RFLP (ninety-seven German Shepherd dogs and one Swiss Shepherd dog previously confirmed as *AA* homozygous), it was possible to identify a clear band pattern in ninety-five individuals. Three German Shepherd dogs failed to produce unequivocal band patterns and their genotypes were not identified using this approach.

From the seven individuals selected for Sanger sequencing in this study (One individual with *AG* genotype, three individuals with *GG* genotype and three individuals that could not be genotyped by PCR-RFLP), it was possible to confirm the genotype only in two individuals: one individual previously identified as heterozygous through PCR-RFLP and one individual that failed in PCR-RFLP analysis and could be identified as *GG* homozygous through sequencing. The electropherogram of the Swiss Shepherd dog previously confirmed as *AA* homozygous via sequencing was used as a reference for the SNP analysis (Fig 2).

In the present study, PCR-RFLP was shown to be an effective method to genotype German Shepherd dogs for the *SOD1* gene (SOD: *c.118G > A*) mutation associated with CDM. Molecular biology techniques, such as PCR-RFLP, may be of clinical use to confirm a diagnosis of genetic disease, and may also be used in screening programs for epidemiological studies [12, 23].

Genetic screening requires precision and a cost-benefit analysis inherent to each location so that it can provide relevant information to breeders on hereditary disorders for genetic counseling, as well as knowledge that can be applied to clinical practice [24]. The present study

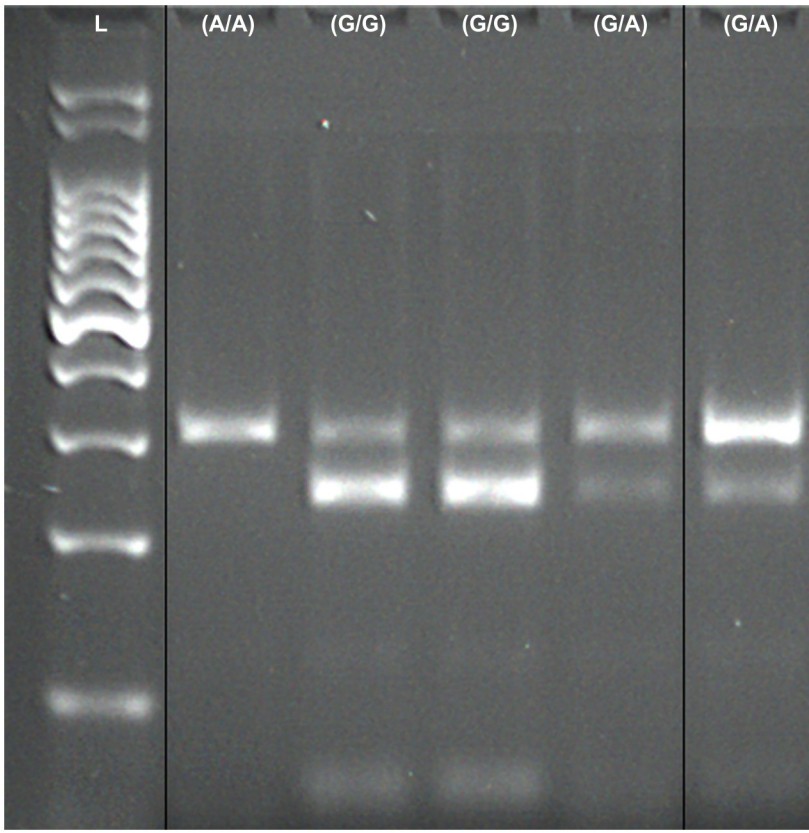

**Fig 1. Electrophoretic (agarose gel) fragment patterns for *SOD1 (SOD: c.118G> A)* mutation associated with CDM in German Shepherd dogs.** The first lane corresponds to a 100bp ladder (Ludwig Biotecnologia LTDA). The second lane corresponds to the Swiss Shepherd dog previously confirmed as *AA* homozygous used as a reference in this study. The third and fourth lanes correspond to *GG* homozygous individuals, and the fifth and sixth lanes correspond to *AG* heterozygous individuals. The original image was spliced to evidence the different genotype patterns; the black vertical lines represent spliced regions. The original image is presented as S1 Fig.

describes a form of low-cost genetic screening when compared to wide range methods. Though not without cost, PCR-RFLP is still a low-cost method which requires a simpler laboratory structure for its execution.

Combining the results of PCR-RFLP and Sanger sequencing, it was possible to identify the genotype of 95 German Shepherd dogs for the mutation of the *SOD1* gene (SOD: *c.118G > A*) associated with CDM, From these, 72 were identified as wild type homozygous "*GG*" and 23 as heterozygous "*AG*". No homozygous individual for the allele associated with CDM (*AA*) was identified in the population analyzed in this study.

A departure from Hardy-Weinberg equilibrium was not observed (p = 0.2940) for the data generated. Observed and expected genotype frequencies (under Hardy-Weinberg equilibrium) are described in Table 2.

The *SOD1:c.118G > A* mutation associated with CDM is widely distributed within the general canine population [11, 12, 23, 25, 26], as can be seen in Table 3. One study identified the presence of the "*A*" allele associated with CDM for the *SOD1*: 118G > A mutation in 124 breeds, but those authors believed that the number could be even greater since a small number of dogs was tested per breed [25].

The frequency of the "*A*" allele seen in German Shepherd dogs in the present study was 0.121, which is comparatively lower than those reported in German Shepherd populations in

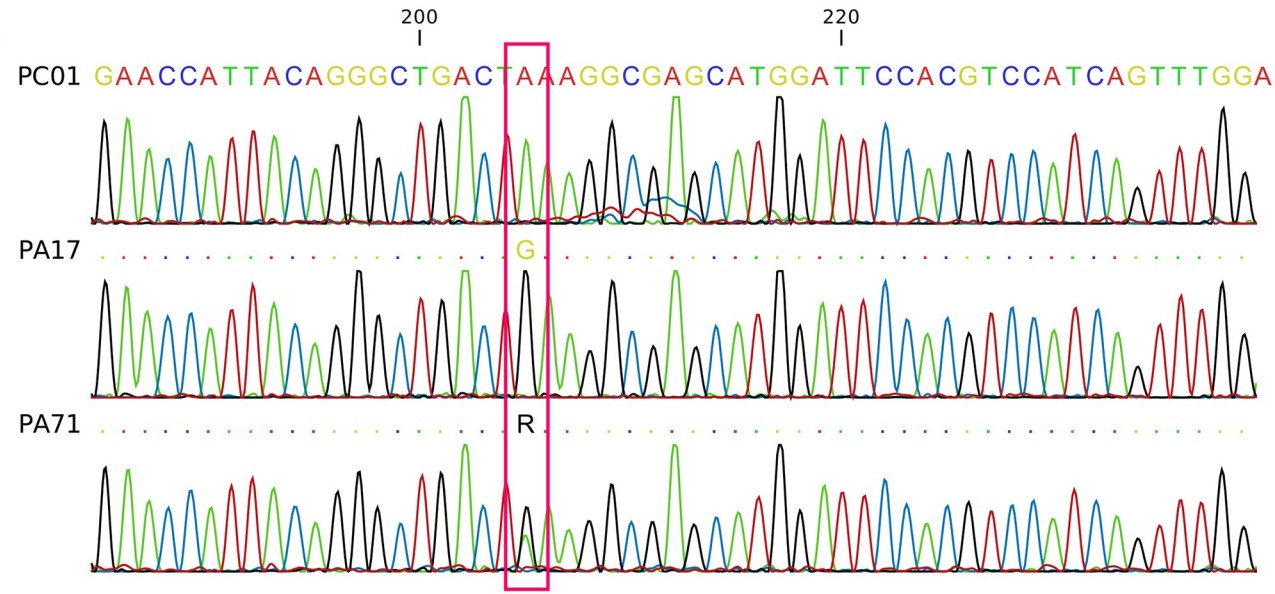

**Fig 2. Electropherograms of the exon 2 of the *SOD1* gene region harboring the *c.118G* > *A* mutation associated with CDM.** A pink rectangle is highlighting the SNP position. The first individual (PC01) is an *AA* homozygous Swiss Shepherd dog used as a reference. The second (PA17) is a *GG* homozygous individual, and the third is a *AG* heterozygous individual.

**Table 2. Genotype and allele frequencies for the mutation of the *SOD1* gene (SOD: *c.118G* > *A*) associated with CDM in German Shepherd dogs.**

| Genotype | Obs[1] | Exp[2] |
|---|---|---|
| *GG* | 0.758 (0.672–0.844) | 0.773 (0.691–0.854) |
| *AG* | 0.242 (0.156–0.328) | 0.213 (0.142–0.283) |
| *AA* | 0.000 (0.000–0.000) | 0.015 (0.003–0.026) |

[1] Observed genotype frequency and its 95% confidence interval;

[2] Expected genotype frequency and its 95% confidence interval.

**Table 3. Reported frequency of the "*A*" allele in breeds predisposed to canine degenerative myelopathy.**

| Breed | n | Frequency of "*A*" | References |
|---|---|---|---|
| Pembroke Welsh Corgi | 3209 | 0.79 | Zeng et al., 2014 [25] |
| Pembroke Welsh Corgi | 122 | 0.7 | Chang et al., 2013 [23] |
| German Shepherd dog | 6458 | 0.37 | Zeng et al., 2014 [25] |
| German Shepherd dog | 150 | 0.38 | Holder et al., 2014 [12] |
| Collie | 151 | 0.39 | Zeng et al., 2014 [25] |
| Collie | 29 | 0.14 | Kohyama et al., 2017 [11] |
| Border Collie | 80 | 0.17 | Zeng et al., 2014 [25] |
| Border Collie | 500 | 0.008 | Mizukami et al., 2013 [26] |
| Bernese Mountain dog | 2413 | 0.38 | Zeng et al., 2014 [25] |

n, number of dogs.

other countries (Table 3). It is important to stress that the genotype and allele frequency obtained in the present study should be analyzed with caution since it was derived from a small group of dogs from a small number of breeders and also from individuals without clinical signs of CDM. As such, the generated estimates may not reflect the real frequency in the entire country. Regardless, the reported frequencies can be considered as the first step in understanding the distribution of the CDM-associated allele in German Shepherd dogs in Brazil.

The presence of the "*A*" allele associated with CDM reported in this study reinforces the need for an initiative to disseminate information related to this disease (and other genetic diseases) in Brazil and paves the way for the construction of a program including researchers, veterinarians, breeders and owners for the detection and control of genetic diseases in Brazilian canine populations. This initiative can certainly contribute to minimize the probability of generating affected individuals [25].

In this sense, a similar investigation has been previously published, where the authors described the clinical relevance of knowing the distribution of the *SOD1*: *c.118G > A* mutant allele in Boxers in South Africa, which allowed clinicians to include CDM as a plausible differential for older dogs with suggestive clinical signs [2].

Therefore, it can be suggested that tests based on DNA analysis can be important when inserted into the diagnostic plan of a patient with clinical signs of CDM. Although CDM is characterized by incomplete penetrance, knowing the genotype of an individual may be particularly useful in determining their potential risk for developing the disease [2, 6].

Knowing the genotype of a young animal with a known mutation may be beneficial for future clinical decisions [27]. A dog that is homozygous for the *SOD1*: *c.118A* allele mutation and has clinical signs compatible with CDM may be presumptively diagnosed with CDM; however, it is important not to neglect other diseases with similar clinical signs that may even occur concurrently with CDM [2, 4]. If that dog has a compressive myelopathy due to intervertebral disc disease confirmed via imaging, decompressive surgery would be the suggested treatment, but it would be known that there is the possibility that he has concurrent CDM [27].

The use of these genetic tests can also be important in genetic improvement programs for breeds known to be predisposed to CDM. A more direct application of DNA testing is to direct breeding decisions, because they can detect carriers of a specific mutation. Decisions based on this information need to be made rationally so that desirable traits and equally important genetic factors are maintained [27].

The first thought may be to eliminate the carriers of a mutant allele from the reproductive population. Although this is a faster way to decrease the frequency of that allele in the population, it may have undesirable consequences, such as the harmful decrease in the gene pool of a breed. Each dog may carry potentially deleterious alleles, and restricting the gene pool may lead to the emergence of a different genetic disease. This may be seen especially in breeds where a mutation has a high allele frequency, for example, CDM in breeds such as Pembroke Welsh Corgi [6, 12, 27].

Dogs that are identified as carriers can still be bred, but with dogs that do not possess the mutant allele. The progeny of this match should be tested because 50% will carry the mutant allele. Genotype should thus be one of the factors that determine which descendants will be used as future breeders. If a carrier dog for the mutant allele has other desirable traits, it can still reproduce, as long as they are not bred with other carriers of that allele. All strategies, if well supported with DNA testing, will contribute toward genetic diversity and promoting ethically desirable characteristics [27].

Due to the complexity of the genetic factor for CDM, and the presence of at least one other causal mutation (the mutation in *SOD 1 c.52T* in exon 1 in Bernese Mountain dogs), care must

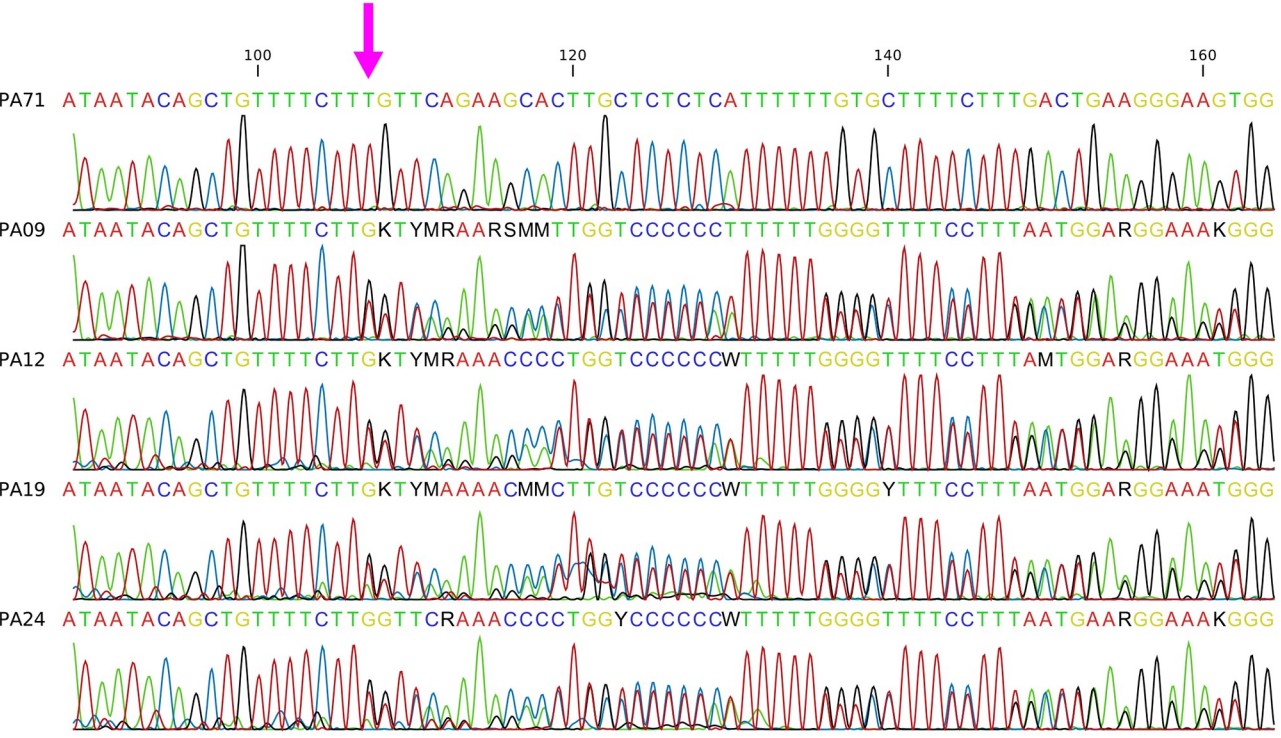

**Fig 3. Electropherograms of the intron 1 of the *SOD1* gene region harboring the newly identified ENSCAFG00000008859:g.26540247del.** A pink arrow is highlighting the deletion position. The first individual (PA71) is homozygous "TT" and used as a reference. The second to fifth individuals (PA09, PA12, PA19, and PA24) are "T-" heterozygous individuals.

be taken when testing a population of dogs for this disease, as well as when developing control and eradication programs [7, 8].

Five individuals did not have sequences of enough quality to allow identification of the genotype for the mutation in exon 2 of the *SOD1* gene (*c.118G > A*). Visual inspection of four of these sequences suggested a possible deletion upstream of the *c.118G > A* mutation, preventing adequate genotyping of the studied polymorphism (Fig 3). One sequence did not generate an electropherogram even after been sequenced again and then was discarded.

To confirm the deletion, Poly Peak Parser software [19] was used to analyze the sequences. Four dogs were heterozygous for a one base pair deletion (S2 Fig). To confirm that the deletion was not an artifact, three samples were sequenced again and the electropherograms showed the same pattern observed previously.

The sequences obtained were compared with a reference sequence (ENSCAFG00000008859) and it can be confirmed as a deletion of one "T" in the position 26540247 described as ENSCAFG00000008859:g.26540247del. Analysis of the ENSEMBL variant database confirmed that the ENSCAFG00000008859:g.26540247del has not been previously reported in dogs (S2 Fig) and that this new mutation is located in the intron 1 of the *SOD1* gene.

Although the role of the ENSCAFG00000008859:g.26540247del on structure and expression of *SOD1* (and consequently its relationship with CDM) was not investigated in this study, its location does not suggest that it can influence the expression of the studied disease.

It is important to highlight that the new described deletion is 94bp far from *SOD1*: *c.118G > A*, and thus, care must be taken in the design of a genotyping procedure. The

PCR-RFLP approach used in this study [12] did not suffer interference from the deletion, since the amplicon produced by the primers did not encompass the ENSCAFG00000008859: g.26540247del. In spite of this, the newly described deletion was responsible for the failure of five out of seven samples sequenced using previously described primers [15].

Aside from economic losses due to failed sample processing, the presence of mutations close to the mutation of interest can also result in genotyping errors, by interfering, for example, in primer annealing causing allelic drop-out [15, 28]. These errors can result in a cascade of erroneous interpretations, favoring the dissemination of a disease-associated allele, which can be particularly harmful with late onset diseases, like CDM [4, 15].

## Conclusions

The present study is the first description of the presence of the "*A*" allele associated with CDM (*SOD1:c.118G > A*) in German Shepherd dogs in Brazil. These results provide important epidemiological data regarding the frequency of this allele in German Shepherds in this country and contribute with the knowledge of the distribution of the aforementioned mutation worldwide.

The PCR-RFLP technique used in this study was well-adapted to Brazil's reality and was considered a useful molecular tool.

And finally, the new described ENSCAFG00000008859:g.26540247del should be considered when designing new genotyping procedures for the *SOD1:c.118G > A* mutation in German Shepherd populations.

## Supporting information

**S1 Fig. Original version of the gel image shown in Fig 1.** Lanes indicated with a red "X" were spliced from the presented figure to evidence the different genotype patterns. The first (top) lane corresponds to a 100bp ladder (Ludwig Biotecnologia LTDA). The fourth (top) lane corresponds to the Swiss Shepherd dog previously confirmed as AA homozygous used as a reference in this study. Fifth and sixty (top) lanes correspond to GG homozygous individuals, and seventh and ninth (top) lanes correspond to AG heterozygous individuals.
(PDF)

**S2 Fig. Partial sequence of the *SOD1* gene indicating the location of the ENSCAFG00000008859:g.26540247del (highlighted in red).** Two previous described mutations are highlighted in blue. Exon 2 region is highlighted with yellow background.
(PDF)

**S1 Table. Signalment, location and genotype of individual dogs enrolled in the present study.**
(DOCX)

## Acknowledgments

The authors would like to thank the breeders who volunteered their dogs for DNA collection and Dr. Felipe Purcell de Araújo and Dr. Italo Barbosa Lemos Lopes who kindly provided an oral swab of a dog with a confirmed case of canine degenerative myelopathy for positive control of the reactions. The authors would also like to thank Dr. Marília de Albuquerque Bonelli for her help with translation of the manuscript.

## Author Contributions

**Conceptualization:** Cássia Regina Oliveira Santos, João José de Simoni Gouveia, Durval Baraúna Júnior.

**Data curation:** Cássia Regina Oliveira Santos, João José de Simoni Gouveia, Joel Fonseca Nogueira, Durval Baraúna Júnior.

**Formal analysis:** João José de Simoni Gouveia, Gisele Veneroni Gouveia, Durval Baraúna Júnior.

**Methodology:** Cássia Regina Oliveira Santos, João José de Simoni Gouveia, Gisele Veneroni Gouveia, Flávia Caroline Moreira Bezerra, Joel Fonseca Nogueira, Durval Baraúna Júnior.

**Project administration:** João José de Simoni Gouveia, Durval Baraúna Júnior.

**Resources:** Cássia Regina Oliveira Santos, João José de Simoni Gouveia, Durval Baraúna Júnior.

**Software:** João José de Simoni Gouveia, Flávia Caroline Moreira Bezerra, Joel Fonseca Nogueira.

**Supervision:** João José de Simoni Gouveia, Gisele Veneroni Gouveia, Durval Baraúna Júnior.

**Validation:** João José de Simoni Gouveia, Gisele Veneroni Gouveia, Flávia Caroline Moreira Bezerra.

**Writing – original draft:** Cássia Regina Oliveira Santos.

**Writing – review & editing:** Cássia Regina Oliveira Santos.

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
