## [Decision Letter · Decision Letter 0]

7 Aug 2020

PONE-D-20-21976

Molecular diagnosis of the mutation associated with canine degenerative myelopathy (SOD1:c.118G > A) in German Shepherd dogs in Brazil

PLOS ONE

Dear Dr. Cássia Regina Oliveira Santos,

Thank you for submitting your manuscript to PLOS ONE. After careful consideration, we feel that it has merit but does not fully meet PLOS ONE’s publication criteria as it currently stands. Therefore, we invite you to submit a revised version of the manuscript that addresses the points raised during the review process.

We look forward to receiving your revised manuscript.

Kind regards,

Shawky M. Aboelhadid, PhD

Academic Editor

PLOS ONE

Journal Requirements:

2. In your Methods section, please provide additional details regarding participant consent from the owners of the animals. In the ethics statement in the Methods and online submission information, please ensure that you have specified (1) whether consent was informed and (2) what type you obtained (for instance, written or verbal). If the need for consent was waived by the ethics committee, please include this information.

Reviewers' comments:

Reviewer's Responses to Questions

**Comments to the Author**

1. Is the manuscript technically sound, and do the data support the conclusions?

Reviewer #1: No

Reviewer #2: Partly

2. Has the statistical analysis been performed appropriately and rigorously? 

Reviewer #1: No

Reviewer #2: No

3. Have the authors made all data underlying the findings in their manuscript fully available?

Reviewer #1: Yes

Reviewer #2: No

4. Is the manuscript presented in an intelligible fashion and written in standard English?

Reviewer #1: No

Reviewer #2: No

5. Review Comments to the Author

Reviewer #1: General comments

The manuscript is interesting and concise. However, the findings are relevant only locally. Some further drawbacks are: 1) the allele and genotype frequencies should be reported toghether with the 95% C.I. 2) the T deletion in intron 1 is quite common and not so relevant unless differently demonstrated. 3) most important! the c.465delT is misleadingly annotated since the deletion is intronic and not in the coding region as the annotation seems to suggest. 4) an indel is not an insertion or deletion but as stated in the HGVS is "deletion/insertion (indel) = a sequence change where one or more nucleotides are replaced by one or more other nucleotides". 5) the manuscript seems arranged as a cut and paste of a previous version since few arguments are commented before being presented and the readibility is hampered 6) lines 279-280: I am not aware of other variants causing disease other than the c.52A>T

Reviewer #2: Question 1: Is the manuscript technically sound and do the data support the conclusions?

This manuscript reports a well thought out and executed genetic analysis in a reasonably large population of dogs. However, it fails to address and discuss one of its main findings, the fact that none of the dogs in the study were homozygous for the mutation associated with the disease.

Additionally, the population of German Shepherd dogs used in this study has not been adequately described. Information on age, sex, where the dogs have come from (ie. people’s pets or veterinary clinics) and any clinical signs of CDM should be included. A statement in the acknowledgements section suggests that the dogs in the study might be being used for breeding purposes. In which case it would be important to know how many breeders participated in the study, how closely related the dogs are, and whether there is a bias towards one sex. All this will greatly influence the genetics of the population and may help to explain the results.

Finally, the indel described in the study could simply be a sequencing error. If the gene had been sequenced in both the forward and reverse directions, instead of only unidirectionally, it might have been possible to determine whether the indel was real or not.

Question 2: Has the statistical analysis been performed appropriately and rigorously?

This manuscript contains no statistical analysis of the data. The authors have calculated expected genotype frequencies using Hardy Weinberg but then failed to do any statistical comparison between that and the actual frequencies. Also, the method used to calculate the expected frequencies has not been reported.

Question 3: Have the authors made all the data underlying the findings in their manuscript fully available?

The authors have only reported the summary of their data. A table containing the genotypes of the individual dogs, and any demographic information, should be included as a supplementary file.

Question 4: Is the manuscript presented in an intelligible fashion and written in standard English?

The presentation and language in the manuscript is generally good. However, the following typographical and grammatical errors need to be addressed.

Line 29 – missing the word gene.

Line 30 – “associated with” not “associated to”.

Lines 98 to 104, 126, 130, 137 to 143 – should be “bp” for base pair not “pb”.

Lines 86 and 112 – should be “40 cycles of denaturation” not “40 denaturation cycles”.

Lines 46, 138, 183, 192, 195, 198 and 273 – no new paragraph is needed.

Line 280 – should be “SOD1: c.52A>T” not “SOD1 c.52T”.

Additional suggestions for improving the manuscript:

1. The introduction is very short and contains limited background information. The paragraph on CDM needs to include a more detailed description of the disease.

2. Figure 1 is unnecessary, the description in lines 125 to 128 is sufficient.

3. The legend for figure 2 needs to describe more clearly what the figure contains. Additionally, where did the White Swiss Shepherd DNA suddenly appear from? I assume this is because no homozygous AA German Shepherds were identified in the study. This needs to be explained in the results/discussion.

4. The results of the genotyping by RFLP (Lines 227 to 231) should be moved above the sequencing data results and discussion about genetic screening (ie. to line 150) so that the results/discussion section flows better.

5. Lines 151 to 153 state “In the present study, PCR-RFLP was an effective form of triage because it allowed genotyping of individuals for mutation of the SOD1 gene (SOD: c.118G> 153 A) associated with CDM”. What sort of triage does this refer to? Since no clinical information was included in the study, this sentence does not make sense.

6. Line 166 should be moved up to line 163 so that the paragraph makes better sense.

7. In Figure 3 it is impossible to see any differences between the traces of the 3 dogs at the location of the SNP. I would suggest the images need to be zoomed in closer to the SNP. Additionally, the legend for this figure should state what genotypes are shown in each of the traces.

8. The inclusion of 3 figures (figures 4,5 and 6) to illustrate the possible indel identified in the study is excessive. A single figure would be sufficient with any other necessary data submitted as a supplementary file.

6. PLOS authors have the option to publish the peer review history of their article (what does this mean?). If published, this will include your full peer review and any attached files.

Reviewer #1: No

Reviewer #2: No

---

## [Author Response · Author response to Decision Letter 0]

21 Oct 2020

Journal requirements

1. The manuscript was completely revised, and necessary alterations were performed to adjust the manuscript to journal requirements

2.Ethical statement was included in the material and methods section. The required additional information related to consent was included.

3. The presented gel figure was altered to adhere to journal guidelines and an original uncropped and unadjusted image was included in the supporting information (S1 Figure)

Reviewer 1: 

 The manuscript was completely revised, and alterations were made to highlight the importance of the study for a wider public.

1. The 95% C.I. estimates for the frequencies were included in the results section.

2. To the best of our knowledge, the deletion described in our study was not described previously. In spite of this, there is another T deletion (rs851394212) already described, located 1bp upstream from the deletion described in our study.

The previous version of the manuscript did not adequately address the discussion related to the description of the new variant identified.

We agree that this variant does not provide evidence that it can influence the expression of the studied disease. Regardless, it was shown that it can interfere in the design of genotyping procedures.

Considering this, we consider that the description of new variants located near causal variants can contribute to a better design of genotyping procedures, avoiding problems like those described in Turba et al (2017).

This section was completely revised and rewritten to adequately describe and discuss the new deletion described in this study.

Turba, et al. Evidence of a genomic insertion in intron 2 of SOD1 causing allelic drop‐out during routine diagnostic testing for canine degenerative myelopathy. Anim Genet. 2017;48: 365-368.

3. To avoid misinterpretation of our results and to adjust it to the HGVS guidelines, we altered the name of the described deletion to ENSCAFG00000008859:g.26540247del. Additionally, the section describing and discussing this new mutation was completely revised and rewritten to highlight its intronic location and to adequately discuss the importance of this mutation.

4. The new described mutation is a deletion and not an indel as indicated in the previous manuscript version. We revised the section describing the new mutation identified and performed the alterations necessary to adjust the nomenclature according to HGVS

5. A deep revision of the manuscript was performed, and it was rewritten to facilitate reading, considering the reviewer’s suggestions.

6. To the best of our knowledge, there are two causal mutations identified for CDM, a transition (c.118G>A) in exon 2 of the SOD1 that was described in several breeds and a transversion (c.52A>T) in exon 1 of the same gene described in Bernese Mountain dogs. Additionally, a haplotype in SP110 gene was described as a modifier related to the time of onset of this disease in homozygous individuals for the A allele in SOD1: c.118G>A. 

In the previous version of the manuscript this paragraph was inadequately presented and has now been completely revised and rewritten. The paragraph included in the new version of the manuscript is: “Due to the complexity of the genetic factor for CDM, and the occurrence of, at list other causal mutation (the mutation in SOD 1 c.52T in exon 1 in Bernese Mountain dogs), care must be taken when testing a population of dogs for this disease, as well as when developing control and eradication programs [6,7].”

Reviewer 2

1. A better description of the studied population was included in the text and in Table 1. The results and discussion section were completely reviewed and rewritten to better address the main findings of the study.

2. A detailed description of the population was included in material and methods section and a Supplementary Table 1 was included with a detailed description of the individuals used in the present study.

3. To confirm that the deletion is not an artifact, three samples were sequenced again and the electropherograms showed the same pattern observed previously. The information about sequencing and analysis of the new identified polymorphism was revised and rewritten to adequately describe and discuss the new deletion described in this study.

4.The information was included in the material and methods section and the 95% C.I. estimates for the frequencies were included in the results section.

5. A detailed description of the population was included in material and methods section and a Supplementary Table 1 was included with a detailed description of the individuals used in the present study.

6. The manuscript was completely revised, and necessary alterations were performed to adjust the manuscript to journal language requirements. All suggestions indicated by the reviewer were addressed in the revised manuscript.

7. The introduction was completely revised and rewritten to include a more detailed description of CDM in all its aspects (clinical, pathological, genetical and epidemiological) and to highlight the importance of the study for a wider public.

8. A revision in the presented figures was performed and the new submitted manuscript kept only three figures. Additional figures were included in the supplementary material.

9. All the legends were revised and rewritten to describe more clearly the presented images. The detailed information of the White Swiss individual was explicitly included in the material and methods section.

The “White Swiss Shepherd DNA” is a DNA sample from a Swiss Shepherd dog previously confirmed as homozygous (A/A) for the CDM associated mutation (SOD1:c.118G > A) and with clinical and pathological signs compatible with the studied disease (Santos et al., 2020) used as a control for the genotyping procedure.

SANTOS, et al. Achados clínicos, histopatológicos e moleculares da mielopatia degenerativa canina: relato de caso [Clinical, histopathological and molecular findings of canine degenerative myelopathy: case report]. Arq. Bras. Med. Vet. Zootec. [online]. 2020;72(2):339-345.

10. A deep revision of the results and discussion section was performed. It was rewritten to improve readability and considering the reviewer’s suggestions.

11. A detailed description of the population was included in material and methods section and a Supplementary Table 1 was included with a detailed description of the individuals used in the present study.

Considering that the focus of the investigation was not to genotype individuals presenting clinical signs of CDM, the paragraph was rewritten to: 

“In the present study, PCR-RFLP was shown to be an effective approach to genotype German Shepherd dogs for the mutation of the SOD1 gene (SOD: c.118G > A) associated with CDM. Molecular biology techniques, such as PCR-RFLP, may be of clinical use to confirm a diagnosis of genetic disease, and may also be used in screening programs for epidemiological studies [9,17].”

12. A deep revision of the results and discussion section was performed. It was rewritten to improve readability and considering the reviewer’s suggestions.

13.This figure was redrawn and one individual homozygous for the wild allele (TT) was included to highlight the mutation in the four heterozygous individuals. A pink arrow was included in the figure to pinpoint the position of the mutation and the figure legend was completely rewritten to better describe the genotype of every individual.

14. The unnecessary figures were removed and only the electropherogram was maintained in the new version of the manuscript. An additional figure indicating the position of the deletion related to other known mutations in this gene was included in the supplementary material (Supplementary Figure 2).

---

## [Decision Letter · Decision Letter 1]

2 Nov 2020

Molecular screening for the mutation associated with canine degenerative myelopathy (SOD1:c.118G > A) in German Shepherd dogs in Brazil

PONE-D-20-21976R1

Dear Dr. Cassia R Oliveira,

We’re pleased to inform you that your manuscript has been judged scientifically suitable for publication and will be formally accepted for publication once it meets all outstanding technical requirements.

Kind regards,

Shawky M. Aboelhadid, PhD

Academic Editor

PLOS ONE

Additional Editor Comments (optional):

Reviewers' comments:

Reviewer's Responses to Questions

**Comments to the Author**

1. If the authors have adequately addressed your comments raised in a previous round of review and you feel that this manuscript is now acceptable for publication, you may indicate that here to bypass the “Comments to the Author” section, enter your conflict of interest statement in the “Confidential to Editor” section, and submit your "Accept" recommendation.

Reviewer #2: All comments have been addressed

2. Is the manuscript technically sound, and do the data support the conclusions?

Reviewer #2: Yes

3. Has the statistical analysis been performed appropriately and rigorously? 

Reviewer #2: Yes

4. Have the authors made all data underlying the findings in their manuscript fully available?

Reviewer #2: Yes

5. Is the manuscript presented in an intelligible fashion and written in standard English?

Reviewer #2: Yes

6. Review Comments to the Author

Reviewer #2: This current version of the manuscript has addressed all the comments I made at the previous round of the review process.

7. PLOS authors have the option to publish the peer review history of their article (what does this mean?). If published, this will include your full peer review and any attached files.

Reviewer #2: No

---

## [Editor Report · Acceptance letter]

5 Nov 2020

PONE-D-20-21976R1 

Molecular screening for the mutation associated with canine degenerative myelopathy (*SOD1:c.118G* > *A*) in German Shepherd dogs in Brazil 

Dear Dr. Santos:

I'm pleased to inform you that your manuscript has been deemed suitable for publication in PLOS ONE. Congratulations! Your manuscript is now with our production department. 

Kind regards, 

on behalf of

Professor Shawky M. Aboelhadid 

Academic Editor

PLOS ONE